# Differential Regulation of *TLE3* in Sertoli Cells of the Testes during Postnatal Development

**DOI:** 10.3390/cells8101156

**Published:** 2019-09-27

**Authors:** Sangho Lee, Hoon Jang, Sohyeon Moon, Ok-Hee Lee, Sujin Lee, Jihyun Lee, Chanhyeok Park, Dong Won Seol, Hyuk Song, Kwonho Hong, Jin-Hoi Kim, Sang Jun Uhm, Dong Ryul Lee, Jeong-Woong Lee, Youngsok Choi

**Affiliations:** 1Department of Biomedical Science, CHA University, Seongnam 13488, Korea; sangho@chamc.co.kr (S.L.); sj091895@naver.com (S.L.); dongwonseol@gmail.com (D.W.S.); drleedr@cha.ac.kr (D.R.L.); 2Functional Genomics Research Center, Korea Research Institute of Bioscience and Biotechnology, Daejeon 34141, Korea; sojiro38@naver.com (H.J.); jwlee@kribb.re.kr (J.-W.L.); 3Department of Stem Cell and Regenerative Biotechnology, Konkuk University, Seoul 05029, Korea; 1004sh.moon@gmail.com (S.M.); lsw46340@naver.com (J.L.); chanhyeok.park3751@gmail.com (C.P.); songh@konkuk.ac.kr (H.S.); hongk@konkuk.ac.kr (K.H.); jhkim541@konkuk.ac.kr (J.-H.K.); 4Department of Animal Science and Biotechnology, Sangji Youngseo College, Wonju 26339, Korea; sjuhm@hanmail.net

**Keywords:** TLE3, Testis, Sertoli cell, spermatogenesis

## Abstract

Spermatogenesis is a process by which haploid cells differentiate from germ cells in the seminiferous tubules of the testes. TLE3, a transcriptional co-regulator that interacts with DNA-binding factors, plays a role in the development of somatic cells. However, no studies have shown its role during germ cell development in the testes. Here, we examined TLE3 expression in the testes during spermatogenesis. TLE3 was highly expressed in mouse testes and was dynamically regulated in different cell types of the seminiferous tubules, spermatogonia, spermatids, and Sertoli cells, but not in the spermatocytes. Interestingly, TLE3 was not detected in Sertoli cells on postnatal day 7 (P7) but was expressed from P10 onward. The microarray analysis showed that the expression of numerous genes changed upon *TLE3* knockdown in a Sertoli cell line TM4. These include 1597 up-regulated genes and 1452 down-regulated genes in *TLE3*-knockdown TM4 cells. Ingenuity Pathway Analysis (IPA) showed that three factors were up-regulated and two genes were down-regulated upon *TLE3* knockdown in TM4 cells. The abnormal expression of the three factors is associated with cellular malfunctions such as abnormal differentiation and Sertoli cell formation. Thus, *TLE3* is differentially expressed in Sertoli cells and plays a crucial role in regulating cell-specific genes involved in the differentiation and formation of Sertoli cells during testicular development.

## 1. Introduction

Spermatogenesis is a process of the formation of sperms from mature spermatozoa in the seminiferous tubules of the testes [1]. In mammals, there are 12 stages of spermatogenesis in seminiferous epithelium of the testicular tubules [2,3]. Each stage can be distinguished by a combination of different types of male germ cells—spermatogonia, spermatocytes, and spermatids [4]. There are two kinds of cell types in the testes—germ cells and somatic cells. Spermatogonial stem cell, one of the germ cells, differentiates through meiosis to form four round spermatids, eventually leading to the formation of spermatozoa and sperm. There are three types of somatic cells—Leydig cells, peritubular myoid cells, and Sertoli cells. Among these, Sertoli cells are located along with germ cells in seminiferous tubules. They play a crucial role in providing nutritive environment during the development and differentiation of germ cells [5,6]. Sertoli cells have a special tight junction, which forms the “blood–testis barrier.” It divides the seminiferous tubules into basal and adluminal compartments [6,7,8]. Additionally, Sertoli cells are in contact with germ cells for communication and to increase their surface area in seminiferous tubules [9]. Although recent studies have identified genes regulating the functions of Sertoli cells in spermatogenesis [3,6,9,10,11,12,13,14,15], additional studies are required to understand the role of Sertoli cells in male reproduction and fertility.

Transducin-like enhancer of split 3 (TLE3) is a member of the Groucho/TLE family and act as a transcriptional co-repressor [16]. Since these factors do not have DNA binding regions, they recruit DNA-binding proteins to regulate expression of the target genes [17]. Several studies have shown that Groucho/TLE proteins play important roles in the development of cells and embryo [18,19,20,21,22]. However, none of the available studies address the role and regulation of TLE3 in the testes. In this study, we investigated TLE3 expression and regulatory targets in Sertoli cells during postnatal testicular development.

## 2. Material and Methods

### 2.1. Animal Management

Male C57BL/6J mice, postnatal day 7 (PD7), PD10, PD14, PD21, and PD42, were purchased from KOATECH (Pyeongtaek, Korea). The mice were housed in a temperature-controlled facility at CHA University under a 12 h light-dark cycle and fed a normal diet and water. All animal experiments were approved by the Institutional Agricultural Animal Care and Use Committee of CHA University (IACUC no. 160018) and performed according to the Guidelines for the Care and Use of Laboratory Animals.

### 2.2. RNA Preparation, RT-PCR, and qRT-PCR

Total RNA was prepared using Trizol reagent (Invitrogen, Waltham, MA, USA) according to the manufacturer’s protocol. The testes of PD7, PD10, PD14, PD21, and PD42 mice were isolated, and total RNA was extracted as described in previous study [23]. The complementary DNA (cDNA) was synthesized using SensiFAST™ cDNA Synthesis Kit (Bioline, Nottington, UK) according to the manufacturer’s protocol. The reverse transcription polymerase chain reaction (RT-PCR) was carried out using Solg™ Taq DNA polymerase (SolGent, Daejeon, Korea). The PCR conditions were as follows: initial denaturation at 95 °C for 5 min, followed by 30–32 cycles of 95 °C for 10 s, 60 °C for 15 s, and 72 °C for 20 s, and extension 72 °C for 10 min. The expression levels were normalized to those of endogenous *Gapdh*. Semi-quantitative real-time RT-PCR (qRT-PCR) was carried out with CFX96 Touch (Bio-Rad Life Sciences, Hercules, CA, USA) using iQ™ SYBR Green Super mix (Bio-Rad Life Sciences). The qRT-PCR conditions were as follows: initial denaturation at 95 °C for 3 min, followed by 45 cycles of 95 °C for 15 s, 60 °C for 15 s, and 72 °C for 20 s. PCR primer sequences are shown in Table 1. The C_T_ value for each gene was determined in the linear phase of the amplification, and the fold change of gene expression was calculated using the 2^−△△CT^ method [24].

### 2.3. Protein Preparation and Western Blot Analysis

A Sertoli cell line, TM4 cells, was cultured using radioimmunoprecipitation assay buffer (RIPA buffer) with protease inhibitor cocktail (Roche, Indianapolis, IN, USA). Total protein was separated by sodium dodecyl sulfate-polyacrylamide gel electrophoresis (SDS-PAGE) and transferred to a polyvinylidene difluoride (PVDF) membrane (Millipore, Burlington, MA, USA). After blocking with 5% non-fat milk in Tris-buffered saline with Tween-20 (TBST), the membrane was incubated with anti-TLE3 antibody (1:3000, #22094-1; Proteintech, Rosemont, IL, USA) at 4 °C overnight. Subsequently, the membrane was incubated with horseradish peroxidase (HRP)-conjugated goat anti-rabbit secondary antibody (1:1000, G21234; Invitrogen) for 1 h at room temperature (RT). The signal was developed using ECL Western Blotting substrate kit (GenDEPO, Katy, TX, USA) and analyzed using ChemiDoc XRS system (Bio-Rad Life Sciences). 

### 2.4. Immunofluorescence Analysis

To determine the localization of TLE3 in the seminiferous tubule during spermatogenesis, each of the testes samples was fixed in 4% paraformaldehyde (PFA) and embedded in paraffin. Tissue sections of 5 μm thickness were placed on the slides and subsequently deparaffinized. The tissue sections were rehydrated and placed in an Epitope Retrieval Steamer (IHC world, Gyeonggi-do, Korea) for antigen retrieval using the Epitope Retrieval Solution System (IHC world) according to the manufacturer’s protocol. After washing in phosphate-buffered saline (PBS), blocking buffer (4% bovine serum albumin [BSA] and 5% goat serum in PBS) was added to the sections, and the slides were incubated in a humidified chamber for 2 h at RT. Then, the sections were stained with the following primary antibodies at 4 °C overnight: anti-TLE3 (1:500, #22094-1; Proteintech), anti-promyelocytic leukaemia zinc finger protein (PLZF) (1:500, sc-28319 AF647; Santa Cruz Biotechnology, Dallas, TX, USA), anti-synaptonemal complex protein 3 (SCP3) (1:200, ab97672; Abcam, Burlington, CA, USA), anti-lectin peanut agglutinin (PNA) (1:400, L-21409; Invitrogen), anti-DEAD-Box helicase 4 (DDX4) (1:500, ab27591; Abcam), and anti-SRY-box9 (SOX9) (1:250, ab76997; Abcam). The slides were then washed three times with PBS and incubated with Alexa Fluor 546 goat anti-rabbit secondary antibody (1:1500; Invitrogen) or Alexa Fluor 488 goat anti-mouse (1:1500, Invitrogen) secondary antibody for 2 h at RT. 

TM4 cells (5 × 10^3^ cells/2 cm^2^) were seeded onto coverslips and fixed with 4% PFA in PBS overnight. Then, the cells were permeabilized using 0.1% Triton X-100 for 10 min, washed three times with 0.1% BSA (Sigma-Aldrich, St. Louis, MO, USA) in PBS, and subsequently blocked with 1% BSA for 40 min. The cells on the slide were stained with the following primary antibodies at RT for 3 h: rabbit polyclonal anti-TLE3 (1:200, Proteintech) and mouse monoclonal anti-alpha tubulin (α-tubulin) (1:100, Abcam). The cells were then incubated for 45 min with the corresponding secondary antibodies: Alexa Fluor 546 goat anti-rabbit (1:1500; Invitrogen) antibody or Alexa Fluor 488 goat anti-mouse antibody (1:1500, Invitrogen). DNA staining was performed using 4′,6-diamidino-2-phenylindole (DAPI, 1:20000, Life Technologies, Carlsbad, CA, USA). Mounting solution (DAKO, Glostrup, Denmark) was applied to the samples prior to covering them with glass coverslips. All images were obtained using a confocal microscope (Leica Microsystems, Wetzlar, Germany) and analyzed using LAS AF imaging software (Leica Microsystems).

### 2.5. Knockdown of TLE3 and Microarray Analysis

TM4 cells were seeded at the density of 1.0 × 10^6^ cells per 3.3 cm^2^ culture dishes. After 24 h, the cells were subjected to transfection for 4 h with either 90 pmole non-targeting or TLE3-specific siRNA duplexes (#SR420092, OriGene Technologies, Rockville, MD, USA) in 2.5 ml Opti-MEM reduced serum medium (Thermo Fisher Scientific, Waltham, MA, USA) and 10 µl RNAiMAX transfection reagent (Invitrogen), according to the manufacturer’s protocol. 

Microarrays for either siCON (siRNA-control) or siTLE3 (siRNA-TLE3) TM4 cells were performed using GeneChip Mouse Gene ST 2.0 (Affymetrix, Santa Clara, CA, USA). The microarray data were analyzed with RMA using Affymetrix default analysis settings and global scaling as a normalization method. The trimmed mean target intensity of each array was arbitrarily set to 100. The normalized and log-transformed intensity values were then analyzed by Biocore company in Korea, using GeneSpring GX 12.5 (Agilent Technologies, Sanata Clara, CA, USA). Fold-change filters included the requirement that the up-regulated or down-regulated genes show at least 1.5-fold changes compared to those of the control groups. Functional annotation of differentially expressed genes (DEGs) and enrichment analysis was performed with DAVID program (v6.8) and gene set enrichment analysis (GSEA, v2.2.4). In addition, prediction for upstream regulator of DEGs, diseases related to DEGs, and functional annotations were performed by Ingenuity Pathway Analysis (Qiagen, Hilden, Germany).

### 2.6. Statistical Analysis

All experiments were repeated at least three times. Quantitative variables are given as ± standard error of the mean (± SEM). The data were analyzed for statistical significance with the Student’s *t*-test, and *p*-values < 0.05 were considered statistically significant.

## 3. Results

### 3.1. Expression of TLE3 in Mouse Tissues Including the Testes

To compare the expression level of TLE family members (TLE1 to TLE6) in mouse tissues, RT-PCR was conducted using various mouse tissue RNAs from the small intestine, stomachs, kidneys, spleens, livers, hearts, brain, lungs, and testes of six-week-old male mice. The results showed that TLE3 and TLE6 were expressed in the testes, whereas other members including TLE1, TLE2, and TLE4 were widely expressed in all tissues (Figure 1a). The qRT-PCR analysis showed that TLE3 expression in the testes was at least 10 times higher than that in the other tissues (Figure 1b). Then, we explored whether TLE3 shows specific expression in seminiferous tubules of the testes. In order to identify each step of the spermatogenesis process, we performed double staining with TLE3 and cell type–specific markers: spermatogonia-specific PLZF, spermatocyte-specific SCP3, spermatid-specific PNA, and Sertoli-specific SOX9. Interestingly, the confocal imaging analysis showed that TLE3 was located in the spermatogonia, spermatids, and Sertoli cells but not in the spermatocytes (Figure 1c). This implies that TLE3 plays a role in spermatogenesis.

### 3.2. Localization and Differential Expression of TLE3 in the Seminiferous Tubule during Testicular Development

To examine the expression level of TLE3 mRNA during testicular development, RT-PCR and qRT-PCR were performed using total RNAs of testes from PD7, PD10, PD14, PD21, and PD42 mice. The results indicated that TLE3 transcripts in the testes increased gradually with postnatal development (Figure 2a,b). To identify the initial day of TLE3 expression during postnatal testicular development, immunofluorescence analysis was conducted with testes from PD7, PD10, PD14, PD21, and PD42 mice. It was found that TLE3 was expressed as early as PD7. However, the imaging analysis indicated that TLE3 was not detected in Sertoli cells at PD7 (Figure 3c). TLE3 started to express in Sertoli cells of PD10 mice, when the spermatogonia enter meiosis. These results indicate that TLE3 plays a regulating role in Sertoli cells during testicular development. 

### 3.3. Gene profiles after TLE3 Knockdown in Sertoli Cells 

To delineate the mechanism of gene regulation by TLE3, we used a mouse Sertoli cell line, TM4 cell (ATCC®CRL-1715). First, we examined the expression level of TLE3 in TM4 cells. The immunofluorescence analysis with anti-TLE3 antibody showed that TLE3 was highly expressed and concentrated in the nucleus of TM4 cells (Figure 3a). Further, to examine the role of TLE3 in regulation of gene expression, siRNA-mediated knockdown of TLE3 was performed in TM4 cells using siTLE3 system. The RT-PCR and western blot analysis showed that TLE3 transcripts and proteins were successfully suppressed in TM4 cells (Figure 3b,c).

Moreover, we conducted microarray analysis using total RNAs of TLE3-knockdown TM4 cells. The TLE3 knockdown group showed significantly increased expression of 1597 genes and decreased expression of 1452 genes compared to that of the control group (Figure 4a,b). The top 10 genes with expression fold change ≥3 are given in Table 2 and Table 3. The analysis of gene ontology using microarray data indicated that TLE3 knockdown activated genes related to translation and RNA processing, whereas it repressed the genes related to transcriptional regulation and immune response (Figure 4c). 

### 3.4. Role of TLE3 in Sertoli Cells during Spermatogenesis

We performed IPA to examine the possibility of occurrence of a human disease or malfunction upon TLE3 knockdown. The results indicated that several diseases and abnormal functions ensue upon TLE3 knockdown. In particular, disorders related to sex development and Sertoli cell formation occurs. Among the genes that are regulated by TLE3, candidate genes related to Sertoli cell metabolism were selected by IPA. Four genes (*FNDC3a*, *GATA4*, *ARID4a*, and *NR0B1*) were up-regulated and two genes (*SOX9* and *HSD17B4*) were down-regulated upon TLE3 knockdown in TM4 cells (Figure 5a). *FNADC3a* and *GATA4* were associated with formation of Sertoli cells and the testes. *SOX9*, *NR0B1*, and *GATA4* played a role in the differentiation of Sertoli cells. qRT-PCR confirmed that *FNDC3a, ARID4a*, and *GATA4* were significantly increased (Figure 5b). Unlike IPA assay, qRT-PCR results indicated that the expression of *HSD17B4* and SOX9 did not change upon TLE3 knockdown in TM4 cells (Figure 5b). However, the overall results showed that efficient regulation of *TLE3* gene in Sertoli cells is vital for cell-specific gene regulation and cellular development during testicular development.

## 4. Discussion

In this study, we revealed differential expression and localization of TLE3 in Sertoli cells during testicular development (Figure 1). The expression of *TLE3* in Sertoli cells begins to appear at postnatal day 10, when male germ cells enter meiosis (Figure 2). In addition, we observed that knockdown of TLE3 in the Sertoli cell line TM4 caused changes in gene expression profiles (Figure 3 and Figure 4). This indicated important roles of TLE3 in the differentiation and development of Sertoli cells (Figure 5).

Among the TLE family members, we found that TLE3 and TLE6 transcripts are highly expressed in the testes (Figure 1a). Unlike TLE3, TLE6 has been reported in developmental and reproductive biology [25,26]. TLE6 plays roles in embryonic development [25]. Bebbere et al. showed that TLE6 is associated with oocyte maturation in sheep [26]. These findings indicate that TLE6 plays an important role in testicular development. However, there is no study highlighting the importance of TLE3 in reproductive tissues such as testes. To the best of our knowledge, this is the first study showing the role of TLE3 in testicular development. Earlier, TLE3 has been studied in various tissues. Villanueva et al. reported that TLE3 has two contradictory functions during transcriptional activity in adipogenesis [27]. TLE3 also regulates metabolism and cell fate in the bone [28,29,30], pancreas [31,32,33], muscle [34], embryonic stem cell [35,36], and cancer [17,27,37,38,39]. It is interesting to note that the relative level of TLE3 in the testes is much higher than that in other tissues. Moreover, TLE3 expression was spatiotemporally regulated in Sertoli cells of the seminiferous tubules during testicular development. Sertoli cells are known as supporting or “nurse” cells that nurture the developing sperm cells through the phases of spermatogenesis and testicular development [3,5,40,41]. Thus, these studies suggest that the TLE3 function is closely related to Sertoli cells.

In the seminiferous tubule, there are two types of cells: germ cells and somatic cells. Germ cells include the spermatogonial stem cells, spermatocytes, spermatids, and spermatozoa. The differentiation of spermatogonia into spermatozoa is a critical event in the testes. However, this process cannot be completed without the support of Sertoli cells. We used cell-specific markers such as PLZF, SCP3, PNA, and SOX9 to identify the cell types and their expression at each developmental stage. The PLZF is spermatogonia-specific marker [42,43,44]. Our results showed the colocalization of TLE3 on PLZF-specific cells, and this is consistent with the result of previous studies that indicate TLE3 affects the maintenance and differentiation of the spermatogonial stem cells [36]. Interestingly, TLE3 was not expressed in SCP3-specific spermatocyte [43], which undergoes meiosis, and this result suggests that TLE3 regulate the division process to prevent indiscriminate meiosis in the spermatogonia. Its expression seems to be associated with the period when Sertoli cell expresses TLE3. This assertion supports the fact that spermatogonia control the function of Sertoli cells [45]. However, further studies are required to understand the resumed expression of TLE3 in the spermatid and to determine its precise mechanism of action on meiosis of the spermatocyte during spermatogenesis. Among the cell-specific markers used for staining, we focused on SOX9-specific Sertoli cells [46,47,48] because Sertoli cells play an important role in spermatogenesis [5] and TLE3 is highly expressed in Sertoli cells. One of the main functions of Sertoli cells is self-renewal of spermatogonial stem cells and regulation of spermatocyte differentiation [41]. On PD10, the spermatogonial stem cells and Sertoli cells are seen in the seminiferous tubules of mouse testes [13,49]. Interestingly, our results also showed that the expression of TLE3 begins to appear in Sertoli cells of PD10 mice (Figure 2c) and remained high post–testicular development. These results demonstrated that TLE3 is dynamically expressed and exhibits Sertoli cell-specificity.

The microarray analysis and IPA allowed for delineating the molecular pathway regulated by TLE3 in Sertoli cells [50]. GO enrichment analysis showed several sets of genes that were involved in negative or positive regulation of transcription (Figure 4c). TLE3 is one of Groucho/TLE family members, which is known to act as transcriptional corepressor or coactivator (reviewed in [51,52]). A recent study showed that TLE3 suppresses target genes of estrogen receptor alpha (ERα) by interacting with FOXA1 in breast cancer MCF-7 cells [17]. Kokabu et al. demonstrated that TLE3 represses the activity of MyoD transcription factor in proliferative satellite cells [34]. Zaret group observed that TLE3 induces glucagon production with Nkx6.1 in β-cells of the pancreas [53]. On the contrary, Villaueva et al. reported that TLE3 acts as a coactivator. They demonstrated that TLE3 selectively recruits and increases the activity of PPARγ in adipose cells [27,54]. From these results, it can be implied that TLE3 acts as a dual-functioning coregulator for Sertoli-specific gene regulation during testicular development. Our IPA results support this inference (Figure 5a). The IPA data revealed several reproduction-related diseases and abnormalities upon TLE3 knockdown, including disorder related to sex and Sertoli cells development. The results also revealed six genes—*FNDC3A*, *ARID4A*, *GATA4*, *NR0B1*, *HSD17B4*, and *SOX9*, whose roles are well characterized in Sertoli cells and germ cells. Among these, only three—*FNDC3A*, *ARID4A*, and *GATA4*—showed significant change in expression upon *TLE3* knockdown. Fibronectin type-III domain-containing protein 3A (*FNDC3A*) are located in the Golgi vesicles and developing acrosome of spermatids and are required for spermatid–Sertoli adhesion during spermatogenesis [3]. Several studies have shown that *FNDC3a*-null mice are sterile owing to the opening of spermatid intercellular bridges and loss of adhesion between the spermatids and Sertoli cells. AT-rich interactive domain 4A (*ARID4A*) is a retinoblastoma (RB)-binding protein. Wu group reported that both *ARID4A* and *ARID4B* act as transcriptional coactivator for androgen receptor and regulates male fertility [10]. *ARID4A*-null mice showed spermatogenic arrest in the meiotic spermatocyte and link post-meiotic haploid spermatids to RB pathway [10]. The GATA binding protein 4 (GATA4) is a well-known marker for Sertoli cells in the testes. GATA4 is involved in regulating Sertoli cell function [11,14]. GATA4*-*deficient mice showed age-dependent loss of fertility as testicular atrophy and decrease of sperm motility [55]. A recent study showed that GATA4 regulates the blood–testis barrier and lactate metabolism [56]. 

Our data imply that TLE3 may be directly or indirectly involved in the regulation of *FNDC3A*, *ARID4A*, and GATA4 in Sertoli cells. Although the magnitude of increase in their expression is not similar to that in a knockout phenotype, we infer that the proper expression of *TLE3* is important for regulating the function of Sertoli cells during spermatogenesis. In other words, TLE3 suppression induces abnormal cellular responses in Sertoli cells.

In conclusion, to our knowledge, this is the first study providing substantial evidence for the role of TLE3 in Sertoli cells. We showed that TLE3 is highly expressed in the testes and is involved in regulation of Sertoli cell-specific genes. Interestingly, we also found that TLE3 expression is spatiotemporally regulated in Sertoli cells during testicular development. Additional studies are required in the future to explore the TLE3 regulatory pathway and its relationship with human disease. This will provide us with a better understanding of the development and role of Sertoli cells.

## Figures and Tables

**Figure 1 cells-08-01156-f001:**
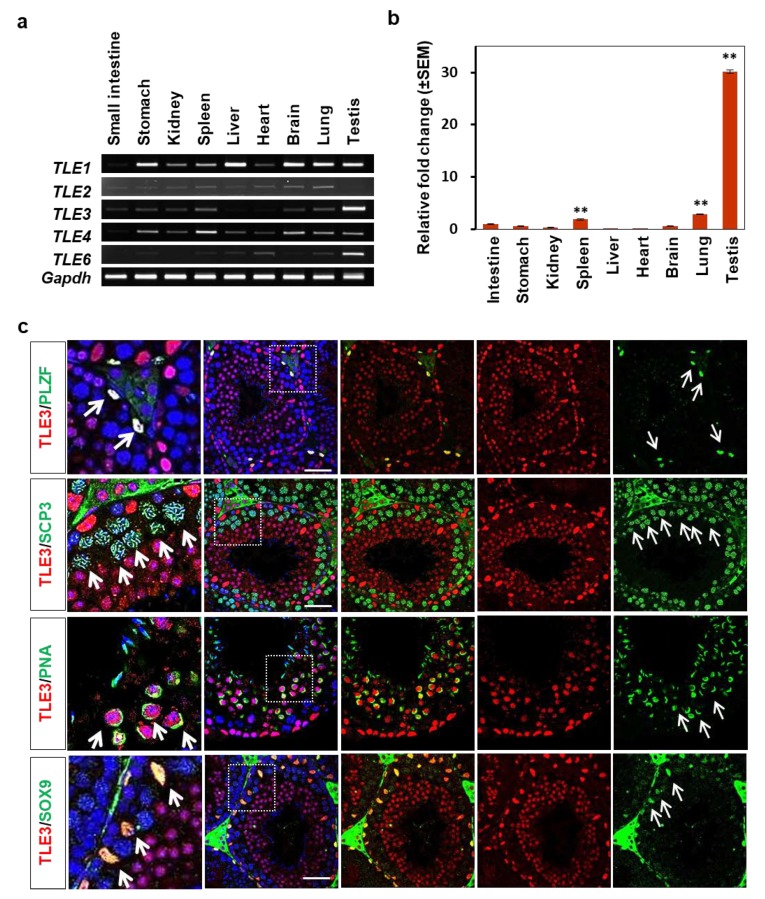
Expression of TLE3 in various tissues and seminiferous tubule in the testes of a mouse. The mRNA was isolated from the tissues of six-week-old male mice. (**a**,**b**) RT-PCR and qRT-PCR analysis of TLE family members (TLE1, TLE2, TLE3, TLE4, and TLE6) in various mouse tissues (intestine, stomach, kidney, spleen, liver, heart, brain, lung, and testis). Relative expression level of TLE3 was normalized with *Gapdh* transcript. Data are represented as mean ± SEM. The Student *t*-test was applied to calculate *p*-value. ***p* < 0.01. (**c**) Immunofluorescence analysis of TLE3 and each stage markers (PLZF, SCP3, PNA, and SOX9) in the seminiferous tubules of the testes of a 6-week-old mouse. Arrows indicate the positive cells with cell-specific antibody. PLZF: spermatogonium marker; SCP3: spermatocyte marker; PNA: acrosome of spermatid marker; SOX9: Sertoli cell marker. DNA was stained with 4′,6-diamidino-2-phenylindole (DAPI). The dotted box with white line represents the magnified region (first column). Scale bar represents 50 μm.

**Figure 2 cells-08-01156-f002:**
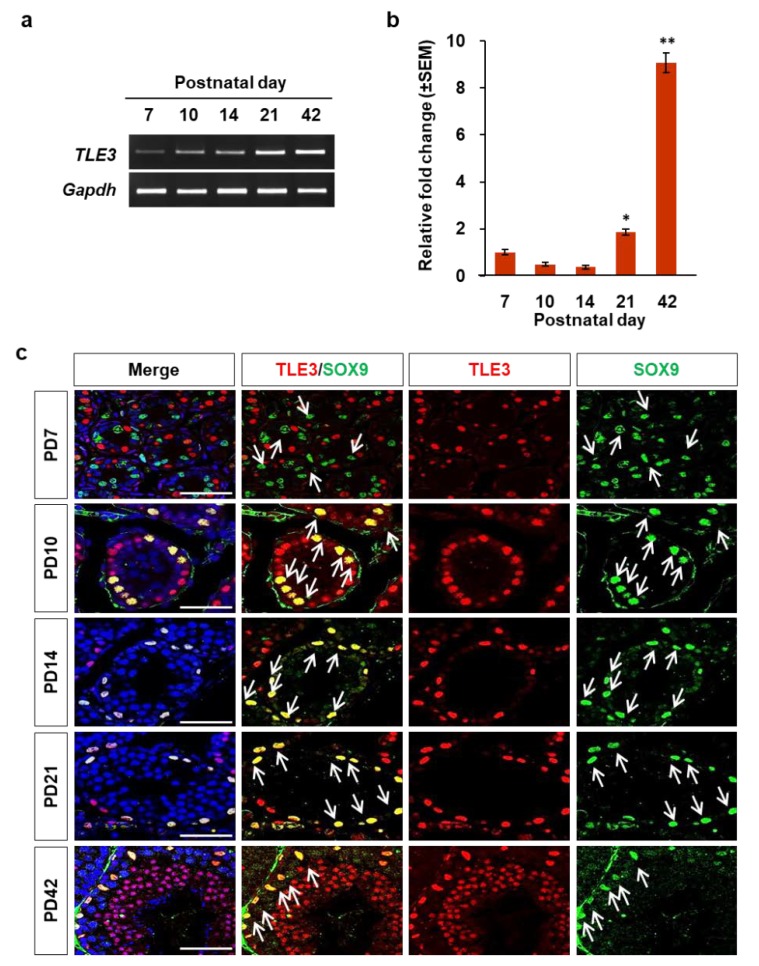
Expression of TLE3 during development of the seminiferous tubule in the testes. The mRNA was isolated from the testes of PD7, PD10, PD14, PD21, and PD42 mice. (**a**,**b**) RT-PCR and qRT-PCR analysis of TLE3 transcript in the testes of PD7, PD10, PD14, PD21, and PD42 mice. TLE3 expression levels were normalized with *Gapdh* mRNA. Data are represented as mean ± SEM. The Student *t*-test was applied to calculate *p*-value. **p* < 0.05, ***p* < 0.01. (**c**) Expression of TLE3 and SOX9 during postnatal testicular development. Nuclei were stained by DAPI. White arrow indicates Sertoli cells. Scale bar represents 50 μm.

**Figure 3 cells-08-01156-f003:**
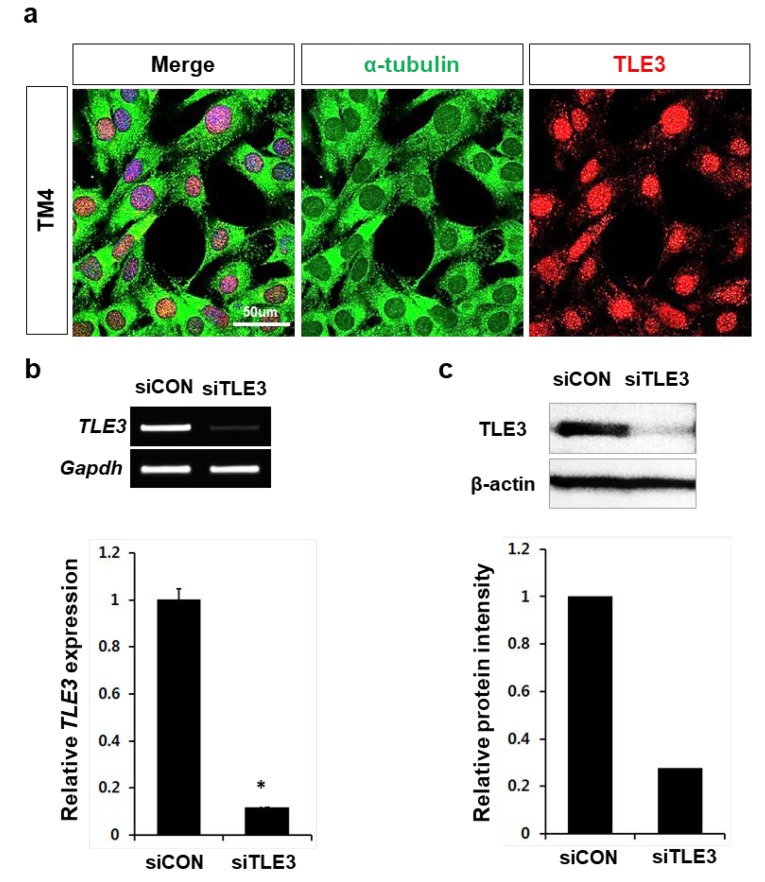
RNAi-mediated knockdown of TLE3 in TM4 cells (**a**) Immunofluorescence analysis of TLE3 in TM4 cells. The alpha-tubulin (α-tubulin) was used as a staining marker of cytosol. Nuclei were stained by DAPI. Scale bar represents 50 μm. (**b**) RT-PCR (upper panel) and qRT-PCR (lower panel) analysis of TLE3 in TLE3*-*knockdown TM4 cells. TLE3 expression levels were normalized with *Gapdh* mRNA. Data are represented as mean ± SEM. The Student *t*-test was applied to calculate *p*-value. ** p* < 0.01. (**c**) Western blot analysis (upper panel) of TLE3 in TLE3*-*knockdown TM4 cells. β-Actin was used as a loading control. The band intensity was analyzed by Image J software (lower panel).

**Figure 4 cells-08-01156-f004:**
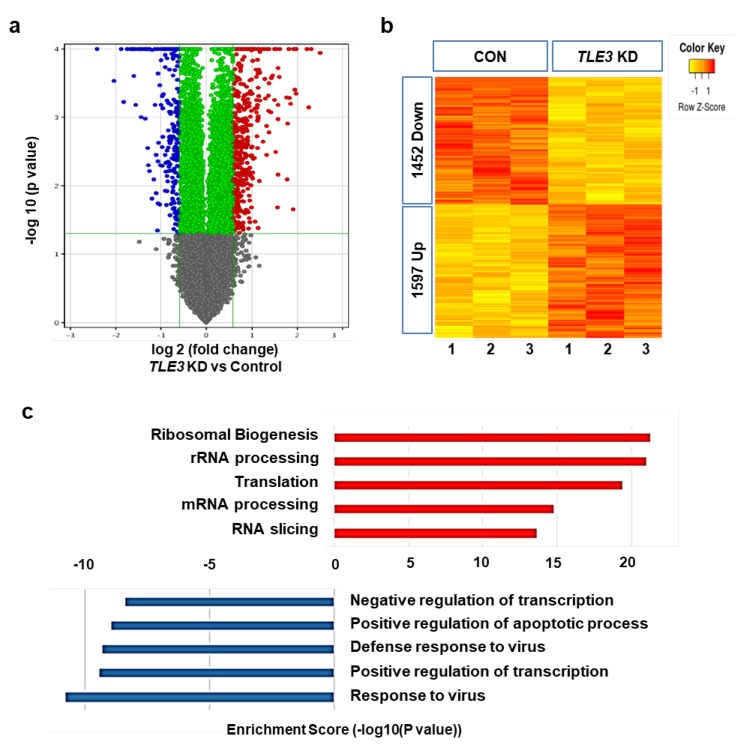
Gene expression profiles upon TLE3 knockdown in TM4 cells. (**a**) Volcano plot analysis between TLE3 knockdown (TLE3 KD) and control (CON). The red point in the plot represents up-regulation, and the blue point represents down-regulation of genes compared to those of CON. The vertical lines correspond to 1.5-fold up- and down-regulation with statistical significance. (**b**) Hierarchical clustering between CON and TLE3 KD. The yellow portion in the graph represents relatively down-regulated genes (1452 Down), and the blue portion represents relatively up-regulated genes (1597 Up) compared to those of the control. (**c**) Gene ontology enrichment and pathway analysis for differentially expressed genes. The score is shown as a fold enrichment value.

**Figure 5 cells-08-01156-f005:**
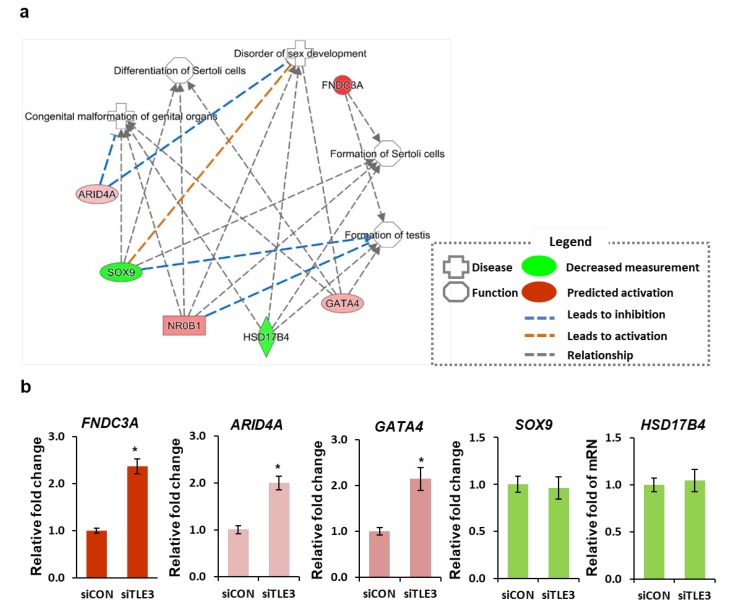
Differential expression of Sertoli cell-associated genes in TLE3-knockdown TM4 cells. (**a**) The gene interaction network for Sertoli cell metabolism generated by Ingenuity Pathway Analysis (IPA). The up-regulated genes are labeled in different shades of red, and down-regulated genes are labeled in green upon TLE3 knockdown. The color intensity represents fold change in gene expression. (**b**) qRT-PCR analysis of candidate genes in TLE3 knockdown TM4 cells. Expression level of different genes was normalized with Gapdh mRNA. Data are represented as mean ± SEM. The Student *t-test* was applied to calculate *p*-value. **p* < 0.05.

**Table 1 cells-08-01156-t001:** Primer sequence for qRT-PCR analysis.

Gene	Forward Sequence	Reverse Sequence
*TLE1*	CCAGTACCTCTCACGCCTCA	GCCCACTCAGAGCACTAGAC
*TLE2*	TGGCTGCCGTAAAGGAAGAC	CTCACTGTCATAAGGCCCTGA
*TLE3*	TCCACCGTCTTTTACCCAAG	CAAGGCCTCATGCAGAGTTA
*TLE4*	TTTACAGGCTCAATACCACAGTC	TGCACAGATAGCATTTAGTCGTT
*TLE6*	ATCCAGTCGGTATTTGTCCATCG	AGGTCTGGGGTTCTACTGAAG
*GATA4*	CCCTACCCAGCCTACATGG	ACATATCGAGATTGGGGTGTCT
*SOX9*	AGTACCCGCATCTGCACAAC	ACGAAGGGTCTCTTCTCGCT
*FNDC3A*	GAACTCCAAAGACGTTGTCCA	GCTTGAATGCAGAACTTGTAGGA
*ARID4A*	GATGAGCCTGCCTACCTGAC	CACCAACTGTGTTGTGTTATCCT
*HSD17B4*	GTCACGGATGACGGAGACTG	GCAAAGCCAAAGCACCAGAG
*Gapdh*	AGGTCGGTGTGAACGGATTTG	TGTAGACCATGTAGTTGAGGTCA

**Table 2 cells-08-01156-t002:** Top 10 up-regulated genes in TLE3 knockdown in TM4 cell.

Gene	GenBank ID	Fold Change	Description
*Mpeg1*	AK030216	5.7	macrophage expressed gene 1
*Pydc3*	AK041099	4.9	pyrin domain containing 3
*Gbp8*	AK041099	4.7	guanylate-binding protein 8
*Saa3*	BC031475	4.6	serum amyloid A 3
*Wfdc17*	BC038870	4.0	WAP four-disulfide core domain 17
*Apod*	AK135046	4.0	apolipoprotein D
*Gm12250*	DQ508486	3.9	predicted gene 12250
*Serping1*	AK144292	3.8	serine (or cysteine) peptidase inhibitor, clade G, member 1
*Gm5431*	DQ508486	3.8	predicted gene 5431
*Angptl4*	AK00916	3.5	angiopoietin-like 4

**Table 3 cells-08-01156-t003:** Top 10 down-regulated genes in TLE3 knockdown in TM4 cell.

Gene	GenBank ID	Fold Change	Description
*Crabp1*	AK045283	−5.3	cellular retinoic acid binding protein I
*Serpinb2*	AK081487	−4.1	serine (or cysteine) peptidase inhibitor, clade B, member 2
*Idi1*	AK029302	−3.6	isopentenyl-diphosphate delta isomerase
*Fdps*	AK077979	−3.4	farnesyl diphosphate synthetase
*Cyp51*	AK002827	−3.1	cytochrome P450, family 51
*Msmo1*	AK005090	−3.1	methylsterol monoxygenase 1
*Sqle*	AK041198	−3.1	squalene epoxidase
*Lss*	AK012813	−3.0	lanosterol synthase
*Msln*	AK144391	−3.0	mesothelin
*Dhcr24*	AK010192	−3.0	24-dehydrocholesterol reductase

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
