# Peer review of "Differential Regulation of TLE3 in Sertoli Cells of the Testes during Postnatal Development"

_cells, 2019, doi:10.3390/cells8101156_

Round 1

Reviewer 1 Report

No additional comments to the reviewer.

Author Response

Thanks for your consideration !

Reviewer 2 Report

Although this study looks interesting, there are major problems rela to results and English writing . Here are major concerns.

1- There is dublicated figure, both of  figure one and two are the same!!!!! Is it accidentally made?

2-  In figure 3, what is MW of TLE3?

3-In figure 4c, what is positive regulation of apoptotic process?

4-in figure 5b, HSD17B4 is upregulated!!!!

5-The English writing is poor. This manuscript should be rewritten by native English speaker.

Author Response

Dear reviewer

We attached a file for the response.

As you  pointed out, we rewrote the manuscript under English editing service.

Best regards,

Youngsok Choi

Round 2

Reviewer 2 Report

The authors improved their manuscript.

Thanks

This manuscript is a resubmission of an earlier submission. The following is a list of the peer review reports and author responses from that submission.

Round 1

Reviewer 1 Report

The involvement of Sertoli cells in the regulation of spermatogenesis is crucial. Understanding and exploring the factors of Sertoli cell origin involved in the development of spermatogonial cells is valuable. Therefore, this study is valuable, but the results are not supporting all the conclusions. The reviewer comments are the following:

Fig. 1C - missing description of the columns. Arrows shoud indicate the type of the stained cells. 

Thre is a staining (green) which is specific for PLZF in the interstitial compartment (Leydig cells and others) Why? 

Fig. 2C needs to add arrows to the stained cells in each column not only on the merged.

The gen expression profile did not show those genes with relation to functionality of Sertoli cells, such as transferrin, androgen binding protein, inhibin B, FSH receptor, and specific growth factors such as GDNF. Those factors are crucial in the development of spermatogenesis and therefore should be presented and discussed.

Those genes should also be confirmed by qPCR analysis.

The possible function of TLE3 in the germ cells was not discussed. 

Reviewer 2 Report

The authors investigated the differential regulation of TLE3 in Sertoli cells of testis during postnatal development. They used mouse model to check the expression of this protein. They knocked down this protein in vitro TM4 cell line and used the microarray analysis to find changes in related genes. By Ingenuity Pathway Analysis (IPA), they found that some changes were upregulated and others were downregulated by such knocking down.

The idea is interesting and the manuscript show novel findings related to TLE3.

There are major problems need to be fixed:

1-The English language needs to be checked by native English speaker, here are examples of unclear English:

In abstract, what is f STK4?

In lines 145 and 146, the authors mentioned that: The semi-quantitative qRT-PCR analysis showed that the expression of TLE3 was at least 10 times higher than that in other tissues.    Where the expression is highest? 

2- The authors should add a few sentences describing the staging of mouse testis in introduction section.

3-In results, Fig.5 legend, the authors mentioned that the up-regulated genes are labeled in red series color and down-regulated genes are labeled
230 in green color by Table 3. Whee is table 3?

also, the histograms in green color (for SOX9 and HSD17B4) did not show any significant differences between control and knocked down TEL3 groups. 

4-In discussion, the authors mentioned in line 303: Our data showed that TLE3 knockdown significantly increased their expression in Sertoli cells!!

Enhanced expression  of which genes?

5- In mouse testis, there 12 stages for spermatogenic cycle, which stage showed the highest expression of TEL3 in Sertoli cells?